# Each Personality Performs Its Own Function: Boldness and Exploration Lead to Differences in the Territoriality of Swimming Crabs (*Portunus trituberculatus*)

**DOI:** 10.3390/biology12060883

**Published:** 2023-06-19

**Authors:** Boshan Zhu, Xin Wang, Ziwen Ren, Hanzun Zhang, Dapeng Liu, Fang Wang

**Affiliations:** 1The Key Laboratory of Mariculture, Ministry of Education, Ocean University of China, Qingdao 266003, China; zhuboshan@stu.ouc.edu.cn (B.Z.); m18660282628_1@163.com (H.Z.); mikeel@126.com (D.L.); 2Function Laboratory for Marine Fisheries Science and Food Production Processes, Qingdao National Laboratory for Marine Science and Technology, Qingdao 266003, China; 3Marine Science Research Institute of Shandong Province (National Oceanographic Center, Qingdao), Qingdao 266104, China; wangxin01@shandong.cn; 4Shandong Yellow River Delta Marine Technology Co., Ltd., Dongying 257000, China; hhsjzhykj@163.com

**Keywords:** *Portunus trituberculatus*, territoriality, personality, behavioral syndrome, marine ranch

## Abstract

**Simple Summary:**

Animal behavior management is an important function of marine ranches, including the personality and the territoriality of animals, with relevant studies having important applications in marine ranch construction. To explore the relationship between the personality and territoriality of swimming crabs, we quantified their boldness and exploration using a behavior observation system. The behavior of crabs with different personalities was measured in a series of environments. The results showed that there was no boldness-exploratory behavioral syndrome in swimming crabs. Territorial behavior was positively correlated with boldness but not with exploration. The difference in boldness determines the behavior response of crabs in a dangerous environment, while exploration difference affects the behavior in habitat selection. Boldness and exploratory behavior explain the difference in the space utilization ability of swimming crabs with different personalities.

**Abstract:**

The boldness and exploration of animals are closely related to their territoriality, with relevant studies having important applications in wildlife conservation. The present study establishes a behavior observation system measuring the boldness and exploration of swimming crabs (*Portunus trituberculatus*) to clarify the relationship between boldness, exploration, and territoriality, as well as to provide a behavioral basis for the construction of marine ranching. The behavioral tests of crabs in a safe environment (predator absence), a dangerous environment (predator presence), and habitat selection (complex and simple habitat) are analyzed. A territorial behavior score is calculated as an evaluation index of territoriality. The correlation between the swimming crabs’ boldness, exploration, and territoriality is analyzed. The results show that there is no boldness-exploratory behavioral syndrome. In predator absence or presence environments, boldness is dominant in territorial behavior and positively correlates with territoriality. Exploration plays a vital role in habitat selection tests but has no significant correlation with territoriality. The experimental results preliminarily show that boldness and exploration jointly develop the difference in the space utilization ability of crabs with different personalities, improving the adaptability of swimming crabs in different conditions. The results of this study supplement the behavior rules of the dominant species of typical fishery resources in marine ranches, providing a basis for achieving animal behavior management function in marine ranches.

## 1. Introduction

Personality is the individual behavioral characteristic of animals that remains consistent across time and context [1]. Personality is usually divided into five major temperament trait categories: boldness, exploration, activity, sociability, and aggressiveness [2]. Among these, boldness and exploration represent behavioral responses to stressors [3]. In some species, such as the killifish (*Rivulus hartii*) [4], cichlid (*Amatitlania siquia*) [5], and green sea turtle (*Chelonia mydas*) [6], boldness and exploration co-evolved commonly positively intercorrelated and formed the boldness-exploration behavioral syndrome [7]. The boldness-exploration behavioral syndrome is the core of individual migration and anti-predation behavior in populations and a fundamental cause of population heterogeneity [8]. However, the boldness and exploration of some species, such as guppies (*Poecilia reticulata*) and parasitic crabs (*Clibanarius symmetricus*), are relatively independent, which do not constitute a behavioral syndrome [9,10]. The boldness and exploration of animals affect their space utilization ability, which is closely related to individual survival and population development and has significant ecological and evolutionary significance [11].

The territorial behavior of animals is a common method of space utilization, including the interaction between animals and their territories and between animals and other individuals within their territories [12]. Personality is an important factor that results in differences in the territorial behavior of animals [11,13]. Individuals with varying levels of boldness have different habitat preferences, territory sizes, and responses to predators and invaders [14,15]. Exploration affects the behavior of animals in unfamiliar environments, their reaction speed when facing novel objects, and their selection of different habitats [16]. In natural habitats, the combination of different personalities enables individuals in the population to have rich behavioral strategies in complex environments and improve the efficiency of food and space utilization, one of the reasons for the formation of territoriality diversification in the population [17]. Exploring the relationship between personality and territoriality helps researchers to understand individual adaptation and behavioral strategy changes more comprehensively. However, most relevant studies focus on terrestrial animals. Therefore, the research on aquatic animals needs to be expanded.

To alleviate the decline of fishery resources caused by high-intensity fishing, China promotes the construction of modern marine ranching, improves the habitat through resource conservation and habitat restoration, and restores aquatic living resources [18]. In marine ranching, the swimming crab (*Portunus trituberculatus*), a typical dominant species of fishery resources in the Western Pacific Ocean, experienced a “cliff” decline due to overfishing that was then followed by gradual resource recovery thanks to marine ranching construction [19,20]. In Laizhou Bay, China, swimming crabs prey on Manila clams (*Ruditapes philippinarum*) but are also the prey of octopuses (*Octopus ocellatus*) (Dan et al., 2019). Behavioral strategies to balance the avoidance of predation risk and resource acquisition are closely related to animal personality [21]. The boldness characteristic of swimming crabs is remarkable [22]. Still, the exploration aspect of their personalities is unclear, and its territoriality is affected by population density [23]. The correlation between personality and territoriality, essential for predicting the population changes and distribution patterns of swimming crabs, needs to be clarified [24]. Therefore, this study establishes an observation system to record and measure the behavior of crabs. According to the results, the boldness and exploration characteristics are calculated. A series of open-field experiments was conducted to determine territoriality. The correlation between personality and territoriality was analyzed. The results of this study supplement the behavior rules of the dominant species of typical fishery resources in marine ranches and provide a basis for improving the animal behavior management function in marine ranches [18].

## 2. Materials and Methods

### 2.1. Animal Collection and Maintenance

The experiment was conducted in the Key Laboratory of Mariculture, Ministry of Education, Ocean University of China, from July to September 2021. Male swimming crabs initiate fighting in most cases because they are bolder and more aggressive than females [25]. Consequently, only male crabs (width = 4.51 ± 0.42 cm, N = 120) were collected from an aquafarm in Huangdao, Shandong Province. Before the experiment, each crab was acclimatized in an independent plastic container (1.68 L, 16 × 14 × 7.5 cm) filled with clean seawater for one week. The numerical symbols in order (1–120) were painted on the carapace as a marker with white acrylic paint for identification. During the acclimatization period, the seawater temperature was 22 ± 1 °C, the salinity was 30 ± 1 ‰, the aeration was maintained continuously, and the photoperiod was 12 L:12 D. Clean mantle and foot muscles of Manila clams were fed to the crabs at 18:00 every day, and the excrements and residual feed were removed at 9:00 the next day. Half of the seawater in the container was changed every day.

### 2.2. Behavior Acquisition and Quantification

The swimming crab behavior observation system was set up indoors, and a series of subsequent open-field experiments were carried out in this system. The system includes a display (PHILIPS, 233I), recorder (Hikvision, DS-7104N-F1, Hong Kong, China), infrared camera (Hikvision, DS-IPC-B12HV2-IA, Hong Kong, China), light source (PHILIPS, 5 W), cylindrical observation box (D = 120 cm; H = 100 cm; Material: PVC), and shade cloth. The infrared camera and light source were located 100 cm above the observation box, whose inner wall and bottom are white. Then, 340 L of clean seawater (Depth = 30 cm) was added to the observation box. During the experiment, there was no aeration in the observation box, and the water temperature and salinity were the same as those during acclimation. The infrared cameras, light sources, and observation boxes were covered with a shade cloth to reduce the interference of external light sources. The experiment was conducted in a quiet, undisturbed room. A total of six identical and independent behavior observation systems were built for simultaneous filming to improve experimental efficiency.

During the experiment, each crab completed four different behavioral tests. Before each part of the experiment, the crabs were starved for 24 h. After the completion of each part, the crabs were moved back to the acclimatizing container. The two adjacent behavioral test experiments were separated across five days to weaken the crab’s memory and improve the environment’s novelty [10]. To avoid the effect of different interval times of two adjacent experiments, each part was completed in the order of numerical symbols. The behaviors of 81 crabs with intact appendages in total were collected. However, 16 crabs molted, and four crabs died during the experiment, with 61 crabs completing all the tests. Only the behavioral data of the crabs that completed all experiments were used in data analysis. The operation time was minimized when changing the water and moving the crabs to reduce the disturbance and protect the welfare of the crabs. After all the experiments were completed, the remaining crabs were returned to the original aquafarm.

#### 2.2.1. Measurement and Classification of Boldness and Exploration

The boldness and exploration measurement of the crabs began at 9:00 every day. The crabs were placed in the box shelter (10 × 8 × 6 cm), and the trapdoor of the shelter (8 × 5 cm) was closed at the beginning (Figure 1A). The crabs were provided with a 10-min adaptation period, and then the trapdoor was opened. After 24 h of continuous recording, the crabs were transferred back to the acclimatizing container. After the simultaneous measurement of each of the six crabs, the observation box and shelter were cleaned and replaced with clean seawater to reduce pheromone residue. Boldness was calculated when the crabs were outside the shelter and during shooting time (h/h) [22,26], while exploration was defined as the time individual crabs were moving within the first 15 min after leaving the shelter (min) [10]. In the previous study, we found that personality of swimming crabs remained stable across repeated tests, thus the boldness and exploration were measured only once [27]. After the shooting, the K-means analysis method was used for clustering. To maximize the measure of the difference of boldness in the crabs, a support vector machine (SVM) was used to calculate the maximum margin hyperplane and classify the boldness and exploration data of crabs. According to the distance between the boldness of the swimming crab and the hyperplane, the crabs were classified using a boldness–shyness dichotomy. Similarly, according to the exploration factor, the crabs were classified according to an explorer–avoidance dichotomy [26].

#### 2.2.2. Behavior Measurement in a Safe Environment

A rectangular shelter (black PVC board, 15 × 10 cm, vertical height = 10 cm) was fixed to the edge of the behavior observation system (Figure 1B), in which the territorial behavior of crabs was recorded. The experiment started at 9:00 every day, and the crabs were put into the system for 24 h, during which the light cycle was the same as the one during the acclimatization period. After 24 h, the behavior of the crabs within 1 h was recorded. The territory size, occupying time (with more than 2/3 of the carapace under the shelter), and defense time (time spent on patrolling) of crabs within 1 h were recorded. Each video captured an image frame every 1 min (a total of 60 frames). Getdata Graph Digitizer 2.26 (WebPlot, Moscow, Russia) was used to capture the coordinates of crabs in the images. All coordinates were used as the original data for analyzing crab territory size and distribution. The kernel method was used to calculate the territory size using Matlab 2019 (Mathworks, Natick, USA). The 95% fixed kernel was used to calculate the territory size, which was then smoothed by the least squares cross value (LSCV).

#### 2.2.3. Predator Response Test

In the predator response test system presented in Section 2.2.2, a cylindrical transparent PVC container (D = 8 cm, H = 20 cm) with holes measuring 1 cm in diameter (eight rows, six holes in each row) was fixed in the observation box at the furthest horizontal distance from the shelter (Figure 1C). An octopus (*O. vulgaris*) (average weight: 210.44 ± 28.12 g) was placed in the container 20 min before the experiment to simulate the predators of swimming crabs in Laizhou Bay. The octopus could observe the crab’s position through the transparent container and extend its head through the hole, but it could not prey on the crab. A crab was then placed in the system, and its behavior within 1 h was recorded. The territory size, occupying time, defense time, and number of times it attacked the predator were recorded during this period. Animal behavior analysis software (Etho Vision XT 10, Noldus Information Technology, Inc., Beijing, China) was used to analyze and map the movements of the crabs.

#### 2.2.4. Habitat Selection Test

Eight square shelters (black PVC board, 10 × 6 × 8 cm, open front and back) were placed in half of the observation box regularly (four shelters were placed along the diameter, and four shelters were placed along the edge of the box), which was considered as a complex habitat (Figure 1D) [28]. The other half area without shelter was considered as a simple habitat, and 3 g of muscle from Manila clams was placed in the center of the simple habitat as prey. A crab was then placed in the system, and the crab’s behavior within 1 h was recorded. The time the crab occupied the complex habitat and the feeding time during this period were recorded.

#### 2.2.5. Territorial Behavior Scores Calculation

The territory size, occupying time, defense time in behavior measurement in a safe environment, the predator response test, feeding time, and time occupying the complex habitat in the habitat selection test were selected and input into the principal component model for dimension reduction analysis. The first principal component (PC1) was used as the “territorial behavior score” to evaluate the territoriality of crabs [28,29,30].

### 2.3. Data Analysis

The data were expressed as mean ± standard deviation (mean ± SD), and *p* < 0.05 was used as the significant difference. The territorial behavior scores were analyzed using principal component analysis. The Spearman’s correlation test was used to analyze the correlation between the boldness and exploration of swimming crabs, as well as the correlation between the boldness, exploration, and territorial behavior scores. The Chi-square test was used to compare the total number of boldness-shyness individuals and explorer-avoidance individuals. The generalized linear mixed model (GLMM) was used to analyze the results of the behavior measurements in a safe environment, predator response test, and territory selection test of individuals with different degrees of boldness and exploration. In the predator response test, the number of times crabs attacked predators was counted only for individuals who attacked the octopus. During the analysis, the territory size, occupying time, defense time in the safe environment, predator response test, feeding time, and time occupying the complex habitat in the habitat selection test were analyzed with skewed distribution; the number of times attacking the predator was analyzed with the negative binomial distribution. Before analysis, the residual of the skewed distribution model was used to check the normal distribution and evaluate whether the binomial distribution model was over-dispersed. Different personalities (based on degrees of boldness and exploration) were used as fixed factors, and the number of each crab (individual ID) was used as a random effect factor to fit the complete model. All data were analyzed using SPSS 24.0 (IBM, New York, NY, USA).

## 3. Results

### 3.1. Clustering Analysis of Boldness and Exploration

Using the results of the K-means cluster analysis, the crabs were divided into groups of either bold or shy individuals according to boldness rating, then divided into groups of either explorer or avoidance individuals according to results of their exploration tests (Figure 2). The boldness of bold individuals was mainly distributed in the range of 0.58–1 (N = 26), while that of shy individuals was in the range of 0–0.58 (N = 35). There was no significant difference in the number of bold and shy individuals (χ^2^ = 1.328, *p* = 0.249). The explorer individuals were mainly distributed in the range of 5.4–15 (N = 24), and the degree of avoidance was mainly distributed in the range of 0–5.4 (N = 37). There was no significant difference in the number of individuals displaying exploration and avoidance (χ^2^ = 2.770, *p* = 0.096). There was no significant correlation between the boldness and exploration of swimming crabs (R = 0.038, *p* = 0.181).

### 3.2. Behavior Measurement in a Safe Environment

There were no significant differences in the territory size (F = 0.010, *p* = 0.921) or occupying time (F = 0.001, *p* = 0.971) between bold and shy individuals. Still, the defense time of bold individuals was significantly higher than that of shy individuals (F = 7.735, *p* = 0.008) (Table 1). There were no significant differences in the territory size (F = 3.134, *p* = 0.082), occupying time (F = 0.508, *p* = 0.478), or defense time (F = 0.277, *p* = 0.601) between individuals displaying exploration and avoidance.

### 3.3. Predator Response Test

In the predator response test, the territory size of bold individuals was significantly more extensive than that of shy individuals (F = 8.625, *p* = 0.035). The occupying time of bold individuals was significantly lower than that of shy individuals (F = 2.427, *p* = 0.012). Still, there was no significant difference in the defense time between the bold and shy individuals (F = 1.170, *p* = 0.284) (Table 2). When facing predators, eight bold individuals (30.8% of the bold individuals) attacked the octopus, while only four shy individuals (11.4% of the shy individuals) attempted to attack the octopus. The results show that the number of times bold individuals attacked the predator was significantly higher than that of shy individuals (F = 2.251, *p* = 0.016). The records of tracks showed that some bold individuals moved several times within reach of the octopus and attacked it (Figure 3A). In contrast, most of the shy individuals concentrated near the shelter and away from the predator (Figure 3B). The analysis of the performance of crabs with different tendencies toward exploration showed that there were no significant differences in territory size (F = 0.873, *p* = 0.354), occupying time (F = 0.001, *p* = 0.992), or defense time (F = 0.259, *p* = 0.613) between explorer and avoidance individuals in the predator response test. When facing predators, seven explorer crabs (29.1% of the explorer individuals) attacked the octopus, and five avoidance crabs (13.5% of the avoidance individuals) attacked the octopus. The results showed that the times the explorer individuals attacked the predator was significantly higher than that of the avoidance individuals (F = 1.034, *p* = 0.033).

### 3.4. Habitat Selection Test

In the habitat selection test, there were no significant differences between the bold and shy individuals in terms of the time spent occupying the complex habitat (F = 0.005, *p* = 0.945) and feeding time (F = 0.731, *p* = 0.396) (Table 3). The time explorer individuals spent occupying the complex habitat was significantly less than that of avoidance individuals (F = 0.773, *p* = 0.018), and there was no significant difference in the feeding time between explorer and avoidance individuals (F = 0.171, *p* = 0.681).

### 3.5. Correlation Analysis of Boldness, Exploration, and Territorial Behavior Scores

Analysis of the results of the behavior measurements in a safe environment, predator response test, and habitat selection test shows that the territorial behavior score (PC1) accounts for 48.24% of the total variance (Table 4). The territorial behavior score was significantly and positively correlated with boldness (R = 0.391, *p* = 0.002) and higher in bold individuals (Figure 4A). There is no significant correlation between territorial behavior score and exploration (R = 0.002, *p* = 0.706). The territorial behavior score of exploration and avoidance individuals were the same (Figure 4B).

## 4. Discussion

The differences in animal personalities lead to the stable differentiation of behavioral strategies and resource utilization patterns of animals within a population [31]. Personality leads to territorial behavior differences by influencing individual predation and anti-predation strategies, as well as the dispersal and migration strategies of populations [28,31]. Similar to cichlids (*Amatitlania nigrofasciata*) [28], differences were found in the behavior of swimming crabs with different boldness and exploration in a series of open-field experiments assessing their territoriality (Table 1, Table 2 and Table 3; Figure 2 and Figure 3). In the safe environment, the defense time of bold individuals was significantly higher than that of shy individuals (Table 1), indicating that they can absorb more time and physiological costs for obtaining and occupying resources, which is similar to the research on fiddler crabs (*Uca mjoebergi*) [32]. In previous studies conducted by the present authors, differences were found in the oxygen consumption rate, energy substance, and relative expression of genes related to energy metabolism between bold and shy swimming crabs, which may be one of the physiological bases for the difference in defense time [27]. In the safe environment, there was no significant difference in territory size between individuals with different levels of boldness (Table 1). Still, when facing a predator, the territory occupied by bold crabs was significantly greater than the shy individuals (Table 2), which is probably because the resource demands of the bold and shy individuals are similar in a stable and risk-free environment. However, in a dangerous environment, bold individuals still occupy a larger territory, exhibiting a “hawk” type strategy that is characterized by high risk and high return [33]. In contrast, shy individuals reduced their territory and increased occupying time to reduce the risk of predation. This particular strategy belongs to the conservative “dove” type [33], which may account for those individuals with different levels of boldness having various strategies and abilities to obtain resources in unfamiliar environments [2]. In the present study, the number of times bold individuals attacked the predator was significantly higher than that of the shy individuals; meanwhile the occupying time was significantly lower (Table 2, Figure 3). The possible reason for this finding is that when facing a predator, some bold individuals thought it was less of a threat and adopted the “direct fighting” strategy, where they approached and attacked the octopus several times. However, the octopus may be a significant threat to the shy individuals, so they instead took the “escape” strategy, where they escaped from the octopus and returned to cover. Some shy individuals considered it so dangerous that they could not escape and resorted to the most extreme “sitting” strategy, stopping their activity to avoid attracting the predator’s attention [34]. Interestingly, two bold individuals tried to prey on the head of the octopus. This behavior occurred most likely because they had not been exposed to predators in the hatchery, leading to their abnormal actions. Thus, dampening susceptibility to predators by properly adding the chemical and visual signals of a predator before releasing might be a way to improve survival in marine ranches and an effective means for stock enhancement of crustacean resources [35].

Boldness affects individuals’ behavior toward potential risk, while exploration reflects individuals’ differences in exploring novel environments and objects [6]. In this study, there was no significant difference in the performance of explorer and avoidance individuals in safe environments (Table 1), indicating that exploration differences did not affect their territorial behavior. When facing predators, explorer individuals attacked the octopus significantly more often than avoidant individuals (Table 2), indicating that exploration affects the exploring behavior on novel objects, which is consistent with the finding that explorer cichlids (*A. nigrofasciata*) had a weak predating-avoidance response to predators [36]. Church [28] pointed out that complex habitats can prevent predators from invading, which limits individual activities. An individual’s preference for habitat complexity reflects their trade-off between risks and benefits, which may be influenced by personality. In the habitat selection test, the differences in time occupying complex habitats (Table 3) demonstrate the correlation between exploration and individual preference for habitat complexity. The difference in individual exploration in the population has profound ecological significance. For example, the explorer Japanese quail (*Coturnix japonica*) shows a higher exploring tendency and makes rapid but brief explorations of unfamiliar areas, which rapidly expands the population distribution. Avoidance individuals have little interest in exploring unfamiliar areas but will slowly and fully exploit the resources in the known areas [37]. Individuals with different exploration survival strategies in the population are complementary, optimizing the resources in the habitat and rapidly expanding the population size [11]. Studies on the red squirrel (*Sciurus vulgaris*) have shown that the evaluation of personality and the collocation of individuals with different personalities can effectively improve the efficiency of population establishment and expansion [38]. Due to imperfect experimental conditions, only the relationship between personality and territoriality of male crabs was measured. Relevant studies, such as the territoriality of crabs in different periods and of different gender, need to be carried out and improved in marine ranch construction.

The existence of syndromes increases the speed of species evolution but limits the plasticity of behaviors and the diversity of personality combinations [30,31]. The present study’s results support no significant correlation between boldness and exploration in swimming crabs, and there was no boldness-exploration behavioral syndrome (Figure 2). In other words, boldness and exploration evolved independently, allowing species to adapt to more habitat types during population formation and migration [15]. The territoriality of the swimming crab was significantly correlated with boldness but not with exploration (Figure 4). However, different amounts of boldness in safe and dangerous environments led to significant differences in individual behavior and played a dominant role in territorial behavior change. In contrast, in habitat selection, exploration had a greater impact (Table 1, Table 2 and Table 3). Therefore, although there is no correlation between boldness and exploration, the characteristics do not constitute a syndrome but jointly explain the differences in space utilization by swimming crabs with different personalities and improve their adaptability in different environments. In marine ranches, swimming crabs may experience multiple environmental changes from the juvenile to adult stages [19]. In subsequent studies, attention should be given to exploring the correlation between personality and territoriality in various environments [11,39].

## 5. Conclusions

There is no boldness-exploration behavioral syndrome in swimming crabs. There is a positive correlation between boldness and territoriality, while no significant correlation exists between exploration and territoriality. Domesticating the identification of predators, evaluating the perfect personalities, and combining different personality are some effective methods that can improve the survival rate of swimming crabs. Personality traits and syndromes are bridges connecting behavioral science, ecology, development, and evolution that are of great significance in exploring the behavior rules of crabs guiding crustacean culture [2,40,41]. In future research, studies should be expanded to explore populations and communities to provide a more systematic behavioral basis for the construction of marine ranching.

## Figures and Tables

**Figure 1 biology-12-00883-f001:**
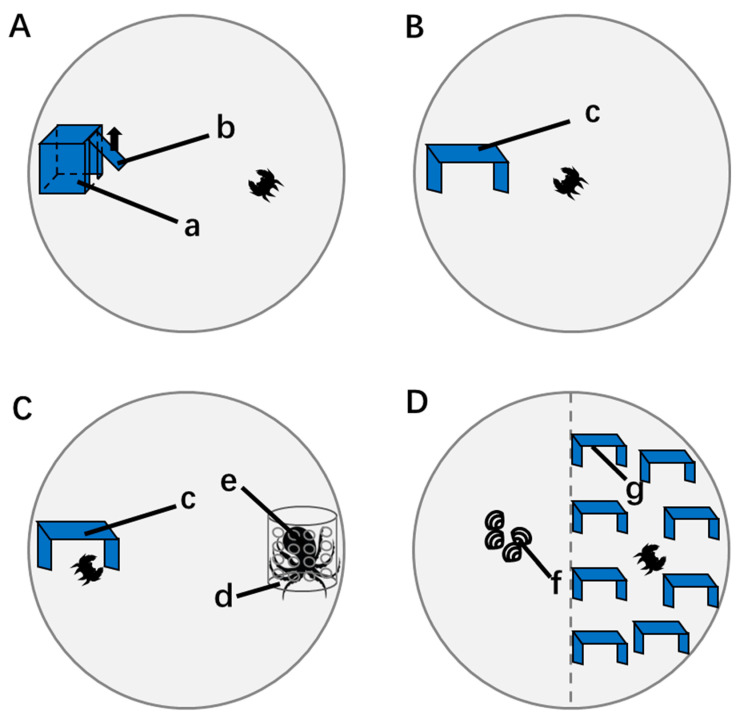
(**A**) System for measuring the boldness and exploration of crabs. **(B)** Behavior measurement in the safe environment, (**C**) predator response test, and (**D**) habitat selection test. a—box shelter, b—trapdoor, c—rectangular shelter, d—cylindrical transparent container, e—octopus, f—square shelter, and g—mantle of clams.

**Figure 2 biology-12-00883-f002:**
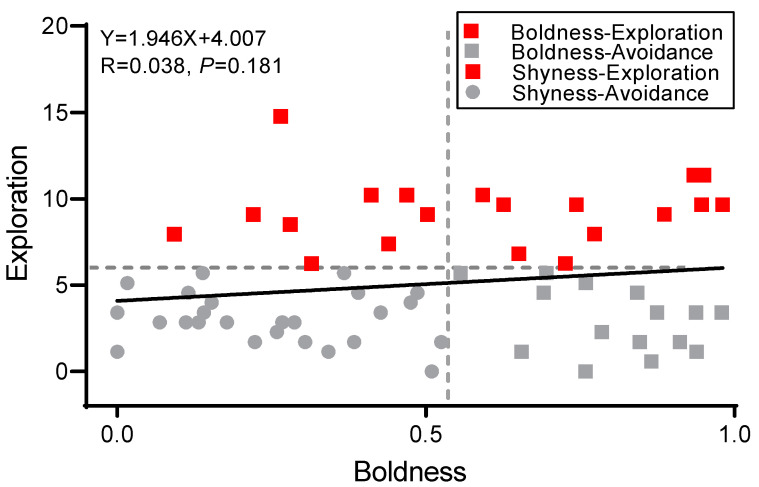
The result of clustering analysis based on boldness and exploration. Notes: The straight line indicates the correlation analysis results of individual boldness and exploration of crabs. The vertical and horizontal dashed lines show maximum-margin hyperplanes according to classification of boldness and exploration respectively.

**Figure 3 biology-12-00883-f003:**
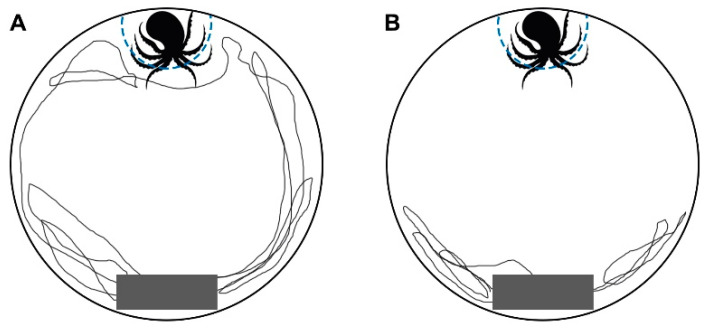
Activity trajectory of the (**A**) bold and (**B**) shy crabs in the predator response test, in which the gray rectangle is the shelter.

**Figure 4 biology-12-00883-f004:**
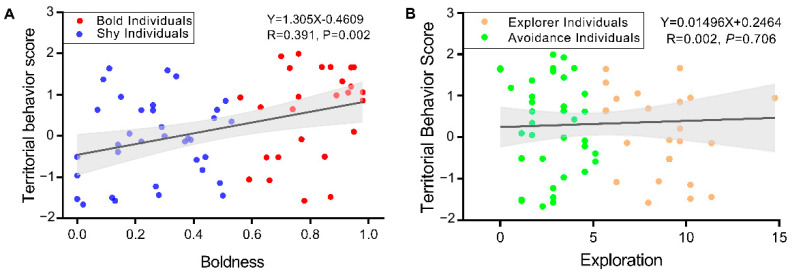
Correlation analysis between (**A**) boldness, (**B**) exploration, and territorial behavior scores of *P. trituberculatus*.

**Table 1 biology-12-00883-t001:** Results of GLMMs analysis of territory size, occupying time, and defense time of individuals with different boldness and exploration of the behavior measurement in safe situations, in which the *p* values in bold represent significant differences among different personality categories (*p* < 0.05).

Behavioral Component	Boldness		*F*	*p*	*df*1, *df*2	AIC	Intercept	Residual
	Boldness Individuals	Shyness Individuals						
Territory size (m^2^)	0.46 ± 0.14	0.46 ± 0.13	0.010	0.921	1, 59	6.187	0.467	0.014
Occupying time (s)	1939 ± 646	1929 ± 468	0.001	0.971	1, 59	404.792	1230.055	2231.553
Defense time (s)	2158 ± 176	1564 ± 838	7.735	0.008	1, 59	314.512	1364.813	872.252
**Behavioral Component**	**Exploration**		** *F* **	** *p* **	***df*1, *df*2**	**AIC**	**Intercept**	**Residual**
	**Explorer** **Individuals**	**Avoidance** **Individuals**						
Territory size (m^2^)	0.53 ± 0.04	0.43 ± 0.13	3.134	0.082	1, 59	9.335	0.433	0.013
Occupying time (s)	2064 ± 222	1869 ± 154	0.508	0.478	1, 59	404.182	1170.358	2212.5
Defense time (s)	1801 ± 169	1692 ± 118	0.277	0.601	1, 59	379.662	993.05	1446.512

**Table 2 biology-12-00883-t002:** Results of GLMMs analysis of territory size, occupying time, defense time, and times attacking the predator of individuals with different degrees of boldness and exploration in the predator response test, where the *p* values in bold represent significant differences among different personality categories (*p* < 0.05).

Behavioral Component	Boldness		*F*	*p*	*df*1, *df*2	AIC	Intercept	Residual
	Boldness Individuals	Shyness Individuals						
Territory size (m^2^)	0.55 ± 0.02	0.43 ± 0.12	8.625	0.035	1, 59	−45.523	0.439	0.007
Occupying time (s)	1672 ± 639	2362 ± 313	2.427	0.012	1, 59	400.621	1263.820	2079.338
Defense time (s)	955 ± 121	781 ± 105	1.170	0.284	1, 59	348.599	689.619	861.242
Times attacking the predator	5.87 ± 1.01	2.25 ± 1.42	2.251	0.016	1, 10	62.84	2.25	2.721
**Behavioral Component**	**Exploration**		** *F* **	** *p* **	***df*1, *df*2**	**AIC**	**Intercept**	**Residual**
	**Explorer** **Individuals**	**Avoidance** **Individuals**						
Territory size (m^2^)	0.51 ± 0.03	0.47 ± 0.02	0.873	0.354	1, 59	−38.443	0.473	0.008
Occupying time (h)	2199 ± 220	2196 ± 153	0.001	0.992	1, 59	402.895	1196.341	2164.897
Defense time (h)	913 ± 139	826 ± 98	0.259	0.613	1, 59	349.395	627.550	874.381
Times attacking the predator	5.25 ± 1.50	3.37 ± 1.06	1.034	0.033	1, 10	63.886	3.375	3.021

**Table 3 biology-12-00883-t003:** Results of GLMMs analysis of time occupying complex habitat and feeding time of individuals with different boldness and exploration in the habitat selection test, in which the *p* values in bold represent significant differences among different personality categories (*p* < 0.05).

Behavioral Component	Boldness		*F*	*p*	*df*1, *df*2	AIC	Intercept	Residual
	Boldness Individuals	Shyness Individuals						
Time occupying complex habitat (s)	2108.70 ± 210.05	2181.00 ± 146.70	0.005	0.945	1, 59	988.317	2181	294.146
Feeding time (s)	301.73 ± 32.29	265.28 ± 27.83	0.731	0.396	1, 59	782.959	265.286	9040.114
**Behavioral component**	**Exploration**		** *F* **	** *p* **	***df*1, *df*2**	**AIC**	**Intercept**	**Residual**
	**Explorer** **Individuals**	**Avoidance** **Individuals**						
Time occupying complex habitat (s)	1664.88 ± 183.04	2277.37 ± 157.76	0.773	0.018	1, 59	987.657	2277.371	290.364
Feeding time (s)	293.35 ± 36.99	274.70 ± 25.84	0.171	0.681	1, 59	783.41	274.707	9125.667

**Table 4 biology-12-00883-t004:** Component loadings for the first (PC1) principal components factor.

Component Loadings	PC1
**Behavior measurement in a safe environment**	
Territory size	0.112
Occupying time	−0.080
Defense time	0.008
**Predator response test**	
Territory size	−0.884
Occupying time	0.967
Defense time	−0.834
**Habitat selection test**	
Feeding time	0.102
Time occupying complex habitat	0.930

## Data Availability

The data presented in this study are available in the article. Further information is available upon request from the corresponding author.

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
