# Peer review of "Each Personality Performs Its Own Function: Boldness and Exploration Lead to Differences in the Territoriality of Swimming Crabs (Portunus trituberculatus)"

_biology, 2023, doi:10.3390/biology12060883_

Round 1
Reviewer 1 Report
This is a high quality, clear, transparent manuscript which provides the reader with confidence in the results presented on the relationship between boldness, exploration, and territoriality in swimming crabs (Portunus trituberculatus) and the consequences for marine ranch construction.
General Comments
The materials and methods section is sufficiently detailed to replicate the experiments. Animal collection, maintenance, acclimation, experimental use and the fate of animals at the conclusion of the experiments (returned to the original aquafarm) are all clearly explained. Weights and/or sizes of animals and equipment used are provided. Figure 1 is a helpful visual explanation of the experimental design.
The experiments were carefully planned (e.g. a total of six identical and independent behavior observation systems were built for simultaneous filming to improve experimental efficiency) and conducted (e.g. to avoid the effect of different interval time of two adjacent experiments, each part was completed in the order of numerical symbols).
The data analysis methods are clearly explained. Whilst P < 0.05 was used as the significant difference, statistical values (e.g. p value, chi-squared statistic, correlation coefficient) are provided to three decimal places, which enables the reader to independently interpret the significance of findings. There is a transparent and logical explanation of which animals were included in the data analysis and why (i.e. only the behavioral data of the (61) crabs that completed all experiments were used in data analysis).
An ethical review statement is provided and the high standard of care provided to the animals used shows an understanding of the importance this has for obtaining quality scientific results.
This is a high standard of English which clearly explains a complex topic. Below are some minor suggestions;
Line 19 ‘ranches’ should be ranch
Line 23 ‘exploratory’ should be exploration
Line 25 ‘Boldness and exploratory explains’ should be ‘Boldness and exploratory behaviour explains’
Line 86 ‘The remarkable boldness characteristic of swimming crabs is remarkable’ write ‘The boldness characteristic of swimming crabs is remarkable’
Line 90 ‘Therefore, this study establishes a behavior observation system. And the behavior of crabs was recorded and measured.’ Suggest writing ‘Therefore, this study establishes an observation system to record and measure the behavior of crabs.’
Line 101 ‘Male swimming crabs, bolder and more aggressive than female, are the initial of fighting in most cases and more typical.’ This sentence does not make sense. Possibly this should be ‘Male swimming crabs initiate fighting in most cases because they are bolder and more aggressive than females. Consequently only male crabs …’
Line 106 ‘The numerical symbols in order (1-120) was painted’ should be ‘The numerical symbols in order (1-120) were painted’
Line 132 ‘experiment’ should be experiments
Reviewer 2 Report
The ms addresses an interesting hypothesis regarding the relationship between boldness, exploration, and territoriality in swimming crabs (Portunus trituberculatus). This appears to be relevant, given the potential applications in wildlife conservation and the potential for this knowledge to aid in the management.
M&M:
A week-long acclimatization period seems reasonable, but the authors may need to justify why this specific duration was chosen and whether it's enough for the crabs to get used to the new conditions.
In 2.2.4, more details about what constitutes a "complex" versus a "simple" habitat could be useful.
It's not clear whether there was a control group in this experiment, or how variables such as crab size, age, or health were controlled.
The Discussion and Conclusion sections could be improved by addressing any limitations of the study.
The ms seems to have some minor language and grammar errors that should be corrected for clarity.
